# Exploiting Chromosomal Instability of PTEN-Deficient Triple-Negative Breast Cancer Cell Lines for the Sensitization Against PARP1 Inhibition in a Replication-Dependent Manner

**DOI:** 10.3390/cancers12102809

**Published:** 2020-09-29

**Authors:** Johanna Rieckhoff, Felix Meyer, Sandra Classen, Alexandra Zielinski, Britta Riepen, Harriet Wikman, Cordula Petersen, Kai Rothkamm, Kerstin Borgmann, Ann Christin Parplys

**Affiliations:** 1Laboratory of Radiobiology & Experimental Radio Oncology, Centre of Oncology, University Medical Center Hamburg-Eppendorf, 20246 Hamburg, Germany; j.rieckhoff@uke.de (J.R.); fe.meyer@uke.de (F.M.); s.classen@uke.de (S.C.); a.zielinski@uke.de (A.Z.); riepen@uke.de (B.R.); k.rothkamm@uke.de (K.R.); borgmann@uke.de (K.B.); 2Department of Tumor Biology, Center of Experimental Medicine, University Medical Center, Hamburg-Eppendorf, 20246 Hamburg, Germany; h.wikman@uke.de; 3Department of Radiotherapy and Radio Oncology, University Medical Center Hamburg-Eppendorf, 20246 Hamburg, Germany; cor.petersen@uke.de

**Keywords:** CIN, TNBC, PTEN, PARP1 inhibition, replication stress, replication fork instability

## Abstract

**Simple Summary:**

The poor prognosis of patients with TNBC have fostered a major effort to identify more patients who would benefit from targeted therapies. Here we recognize *PTEN* as a potential CIN-causing gene in TNBC and consider PTEN-deficient TNBC for the treatment with PARP1 inhibitors due to the protective role of PTEN during DNA replication.

**Abstract:**

Chromosomal instability (CIN) is an emerging hallmark of cancer and its role in therapeutic responses has been increasingly attracting the attention of the research community. To target the vulnerability of tumors with high CIN, it is important to identify the genes and mechanisms involved in the maintenance of CIN. In our work, we recognize the tumor suppressor gene Phosphatase and Tensin homolog (*PTEN)* as a potential gene causing CIN in triple-negative breast cancer (TNBC) and show that TNBC with low expression levels of PTEN can be sensitized for the treatment with poly-(ADP-ribose)-polymerase 1 (PARP1) inhibitors, independent of Breast Cancer (BRCA) mutations or a BRCA-like phenotype. In silico analysis of mRNA expression data from 200 TNBC patients revealed low expression of PTEN in tumors with a high CIN70 score. Western blot analysis of TNBC cell lines confirm lower protein expression of PTEN compared to non TNBC cell lines. Further, PTEN-deficient cell lines showed cellular sensitivity towards PARP1 inhibition treatment. DNA fiber assays and examination of chromatin bound protein fractions indicate a protective role of PTEN at stalled replication forks. In this study, we recognize *PTEN* as a potential CIN-causing gene in TNBC and identify its important role in the replication processes.

## 1. Introduction

Chromosomal instability (CIN) is a type of genomic instability that is defined by the loss or rearrangement of chromosomes that classifies numerical CIN or structural CIN [1]. Defects in several cellular processes, including cell-cycle checkpoint controls, chromosome segregation, DNA repair and DNA replication are known to cause CIN [2,3,4]. The spectrum of gene alterations and mutations that cause CIN is only partially known, but insights would be beneficial for the treatment of tumors with high CIN (CIN^+^).

Phosphatase and Tensin homolog (*PTEN*) is an important tumor-suppressor gene frequently mutated or deleted in human cancer. Loss of PTEN has been associated with aneuploidy and poor prognosis in cancer patients [5]. PTEN is known to play an important role in antagonizing the PI3K-AKT pathway in the cytoplasm [6]. Nuclear PTEN is involved in genome maintenance pathways [5]. Several reports indicated, however, that reduced levels or deletion of PTEN also are associated with decreased homologous recombination (HR) efficiency [6,7,8,9]. Furthermore, PTEN deficiency induces DNA replication stress, disrupts mitotic spindle architecture and leads to the accumulation of structural and numerical CIN. PTEN is a well-known guardian of the genome due to its control of multiple processes maintaining CIN [5].

*PTEN* is also often found to be mutated in triple-negative breast cancer (TNBC) [10], the most aggressive subtype of breast carcinoma. It has been reported that loss of PTEN is frequent in TNBC [11] and is associated with an especially aggressive behavior [12] and advanced stage of cancer such as brain metastases [13]. The lack of estrogen, progesterone and human epidermal growth factor receptor 2 (HER-2) of TNBC leaves little scope for targeted therapies [14]. The inhibition of poly-(ADP-ribose)-polymerase 1 (PARP1) is a promising approach for TNBC with a defined defect in the HR repair pathway based on the concept of “synthetic lethality” [15].

Although PARP1 inhibitor therapy has predominantly targeted Breast Cancer (BRCA)-mutated cancers, there is growing evidence that PARP1 inhibitor treatment for non-BRCA-mutant tumors might be beneficial for CIN^+^ tumors [16,17].

We aimed to determine the role of PTEN in causing CIN in TNBC. First, we performed an in silico analysis of PTEN messenger RNA expression (mRNA) in TNBC and analyzed PTEN expression in different breast cancer cell lines by Western blot. To further explore the mechanism of PTEN in TNBC, we analyzed DNA replication processes by DNA fiber assays and examined the role of PTEN at the chromatin. To measure the sensitivity towards PARP1 inhibition treatment, we performed a cell survival assay.

## 2. Results

### 2.1. Low PTEN Expression Levels Correlate with CIN

To investigate whether PTEN plays a role in the maintenance of CIN in TNBC, an in silico analysis of mRNA expression data from TNBC tumors, previously determined in the Metabric study, was performed [18]. The CIN70 expression signature was derived from a surrogate measure of CIN and is defined as the average expression of 70 genes that correlate with “total functional aneuploidy” in solid tumors [19]. Strikingly, the analysis revealed a significantly lower mRNA expression of PTEN in TNBC with high CIN70 scores (−1.9 ± 0.2) in comparison with TNBC with low CIN70 scores (−0.9 ± 0.13) (Figure 1a), indicating CIN^+^ in TNBC with a low PTEN expression. To further investigate the role of PTEN in vitro, we selected five TNBC cell lines (MDA231, BT549, HS578, BT20, GI101) with no mutation in BRCA1-, BRCA2- or BRCA-like phenotypes and four non-TNBC cell lines (SKBR3, MCF7, T47D, BT474) to analyze PTEN expression. Western blot data showed no expression of PTEN in BT549, intermediate expression in MDA231 and low expression among the three other TNBC cell lines (Figure 1b). On average, TNBC cell lines showed a 1.85-fold lower expression of PTEN with 1.6 ± 0.4 compared to 2.9 ± 0.2 in non-TNBC (Figure 1c).

### 2.2. Low Expression of PTEN Causes Replication Stress in TNBC Cell Lines

Recently, we determined the HR competence of the cell lines investigated here and we could show that HR was reduced in TNBC cell lines [20]. Since CIN often occurs as a result of defective DNA repair and replication stress [21,22], we performed the DNA fiber assay to analyze the performance of PTEN-deficient cells in DNA replication. Exponentially growing breast cancer cell lines were sequentially pulse-labeled for 30 min each with 5-Chloro-2′-deoxyuridine (CldU) and Iododeoxyuridine (IdU). For replication fork progression, the lengths of DNA fibers were measured in untreated controls (CldU- and IdU-labeled fibers) and active replication origins were counted by quantification of IdU-labeled, CldU-negative fibers. While replication fork progression did not differ between TNBC and non-TNBC cell lines (Figure 2a), the firing of second pulse origins was significantly higher in TNBC, with 23.12% ± 3.2, compared to 13.17% ± 2.6 in non-TNBC cell lines (Figure 2b).

### 2.3. Replication-Dependent Sensitization by PARP1 Inhibition Due to Reduced Replication Fork Elongation and Fork Stalling

Inhibition of PARP1 is an emerging strategy that can be used to selectively target genomic unstable tumors [17]. To investigate the effect of PARP1 inhibition on the replication processes of TNBC, we used the DNA fiber assay to analyze the impact of PARP1 inhibition on the elongation processes, replication fork stalling and the activation of second pulse origins (Figure 3a–g). Exponentially growing TNBC cell lines were sequentially pulse-labeled for 30 min each with CldU and IdU. To analyze the effect of PARP1 inhibition on replication processes, 1 µM Olaparib was added 120 min before pulse labelling. For fork progression, the lengths of DNA fibers were measured in treated samples and compared to untreated controls (CldU- and IdU-labeled fibers) (Figure 3b). Fork stalling was analyzed by counting CldU-labeled, IdU-negative fibers (Figure 3c). Newly activated replication origins were counted by the quantification of IdU-labeled, CldU-negative fibers (Figure 3d). As shown in Figure 3e, single treatment with 1 µM Olaparib provoked shorter replication tracts in all TNBC cell lines. Furthermore, replication fork stalling was observed for almost all cell lines, albeit to a lesser degree (Figure 3f). The analysis of replication origins showed that origin firing was downregulated in almost all cell lines after PARP1 inhibition (Figure 3g). However, in contrast to the other cell lines, we could observe that the BT549 cell line with no expression of PTEN showed a strong activation of second pulse origin firing. This may be due to the high extent of replication fork stalling.

Given the HR repair defect and the replication-dependent sensitivity of PTEN-deficient cells, we tested the cellular sensitivity of these cells to Olaparib. For three out of five TNBC cell lines, the IC50 was at or below 1 µM Olaparib and only 10% or fewer of the cells survived the treatment with 10 µM Olaparib (Figure 4).

### 2.4. Lack of PTEN Leads to Replication Fork Instability in TNBC Cell Lines

To further analyze the high amount of replication fork stalling after PARP1 inhibition, we examined the stability of replication forks after treatment with Hydroxyurea (HU) for the TNBC cell lines. Earlier works had shown that the nascent DNA strands of HU-induced stalled replication forks undergo extensive nucleolytic degradation in HR-deficient cells [23]. Exponentially growing TNBC cell lines were pulse-labeled with CldU. CldU was washed out and cells were treated for four hours with 2 mM HU. After HU treatment, cells were washed with phosphate-buffered saline (PBS) and labeled with IdU (Figure 5a). We assessed the stability of replication forks after HU treatment by measuring the length of CldU tracks. TNBC cell lines showed a shortening of the CldU tracks after HU treatment compared to untreated controls, indicating replication fork instability (Figure 5b). BT549 cells with no expression of PTEN showed the strongest reduction in CldU track length (Figure 5c). In addition, we could demonstrate that the amount of degraded DNA negatively correlated with the amount of PTEN (Figure 5d). These data support the idea that PTEN could be a protective factor at replication forks. By further analyzing SKBR3, MCF7, T47D and BT474 with a high amount of PTEN, we could show that PTEN is a general protective factor at stalled replication forks. No change or increase in CldU track length was observed in three out of four cell lines with a high amount of PTEN (Appendix A).

### 2.5. PTEN Protects Replication Forks by Supporting the Recruitment of Repair Proteins to the Chromatin

To study the effects of different protein amounts of PTEN, we selected MDA-MB-231 BR (Brain-seeking) cells to analyze the effects of PTEN overexpression compared to BT549 and MDA231. The MDA-MB-231 BR cell line was established from the parental MDA-MD-231 cell line following serial in vivo passaging to the brain [24] and expresses a lower amount of PTEN compared to MDA231 [25]. MDA-MB-231 BR cells were transduced by lentiviral particles containing the PTEN coding sequence (pPTENiZs2puro++tTR+) under the control of doxycycline response element (MDA231/PTEN). MDA 231/PTEN shows the highest expression of PTEN with 5.4 ± 0.2 (normalised on loading control) compared to 2.6 ± 0.3 for MDA231 with an intermediate expression and no expression of PTEN in BT549 (Figure 6a,b and Appendix A). For MDA231/PTEN and MDA231, we could show that PTEN was located at the chromatin with a 2.2-fold higher expression of PTEN for MDA231/PTEN. Additionally, we observed that BT549 had lower amounts of Partner and localiser of BRCA2 (PALB2), PARP1 and Checkpoint kinase 1 (CHK1) at the chromatin compared with MDA231/PTEN and MDA231 (PALB2: 1.1 vs. 3.4 and 1.8, PARP1 3.0 vs. 16.4 and 8.0 and CHK1 1.1 vs. 4.3 and 4.1) (Figure 6a and Appendix A ). These data lead us to analyze the phosphorylation of Chk1, an important activator of the Intra-S-phase checkpoint after induced replication stress. After HU treatment, we found that there was an almost two-fold lower phosphorylation of CHK1 in BT549 compared to MDA231/PTEN and MDA231 (Figure 6b). Interestingly, the recruitment and activation of the upstream Kinase Ataxia teleangiectasia and Rad3 related (ATR) of the Intra-S-Phase was not affected after HU treatment. Rather, the data show the strongest activation of ATR in BT549 compared to the other cell lines. Finally, we elucidated the effect of different amounts of PTEN on replication stress. We compared the effect of low PTEN expression in TNBC cell lines with TNBC cell line MDA231/PTEN. MDA231/PTEN displayed a faster replication fork progression, with 1.04 kb/min compared to 0.74 kb/min, and, with 19% compared to 24%, a lower amount of origin firing compared to TNBC with low amounts of PTEN in the unperturbed state (Appendix A). HU treatment had a mild effect on CldU track length (Appendix A). After PARP1 inhibition, the elongation rate (0.94) and origin firing (1) did not change compared to the untreated state (Appendix A). Only a slight increase in fork stalling could be detected (Appendix A).

## 3. Discussion

The poor prognosis and molecular heterogeneity of patients with TNBC have fostered a major effort to identify more patients who would benefit from targeted therapies. The aim of our work was to investigate the underlying mechanism of PTEN in the maintenance of CIN and to evaluate the potential of PARP1 inhibitor treatment in TNBC with high CIN.

*PTEN* is mutated in a wide variety of solid tumors. It was reported earlier that disruption of *PTEN* in Mouse Embryonic Fibroblasts (MEFs) leads to extensive centromere breakage and chromosomal translocations due to physical interaction with centromeres and its role in DNA repair [6]. Here, we show that low mRNA expression levels of *PTEN* in tumors from TNBC patients are significantly associated with high CIN (Figure 1a). In vitro analysis aggregates these findings. TNBC cell lines express lower protein levels of PTEN compared to non-TNBC cell lines (Figure 1b). In line with the observation in Prostate cancer cell lines [9], our data provide strong evidence for the involvement of PTEN in double-strand breaks (DSB) repair via HR in TNBC, based on experiments with the reporter plasmid assay [20].

The relationships between HR and DNA replication are well documented [26]. Cells that lack efficient DNA repair systems, particularly HR, show replication stress in the unperturbed state, which is defined as the slowing or stalling of replication fork progression [27]. With this in mind, we were further interested to see if TNBC cell lines with low protein expression of PTEN exhibit replication stress. We demonstrate that all analyzed breast cancer cell lines show slow fork progression (Figure 2a). Moreover, in TNBC cell lines the activation of replication origins was much higher compared to non-TNBC cell lines (Figure 2b). Earlier, we reported that HR-deficiency due to *PALB2*-haploinsufficiency, Rad51AP1-, NUCKS- knockdown and Rad51 overexpression resulted in replication stress due to compromised replication fork progression and high activation of dormant origins [20,28,29,30]. We reveal from our data that PTEN-deficient TNBC cell lines display endogenous replication stress due to uncontrolled origin firing.

We assume that consecutive rounds of replication stress are a strong driver of CIN and not only do they repair deficiency, but endogenous replication stress could make TNBC cell lines eligible for PARP1 treatment. The sensitization of HR-deficient tumors to PARP1 inhibitors was primarily explained by the concept of synthetic lethality [31]. However, the molecular mechanisms that drive this synthetic lethality remain unclear. PARP1 and HR proteins intersect at DSB repair and replication forks [32]. We reported that PARP1 inhibitors enhance the therapeutic ratio achieved by radiotherapy by interfering with the replication elongation for the HR-deficient head and neck squamous cell carcinoma (HNSCC) cell lines [33]. Here, we demonstrate that replication stress in PTEN-deficient TNBC is increased during the inhibition of PARP1, as evidenced by a further reduction in replication fork progression and a high number of stalled replication forks (Figure 3d,e). We observed that BT549 with no expression of PTEN displayed the strongest effects in replication fork progression and fork stalling. In addition, BT549 showed a strong activation of second pulse origin firing upon PARP1 inhibition (Figure 3g). PARP1 inhibitors cause an increase in DNA single-strand breaks (SSBs), which are converted during replication to DSBs in BRCA1/2-defective cells. However, researchers previously primarily studied DNA damage induction, repair capacity and cell cycle progression upon PARP1 inhibition [34,35]. Recent findings have suggested that the slowing, stalling or collapse of DNA replication forks is the prevalent source of DNA damage, which may contribute to genome instability and lead to the generation of structural and numerical CIN [3]. Our work provides evidence that PTEN plays an important role during replication processes. Although previous studies have focused exclusively on replication fork progression and linked slower fork progression with structural and numerical CIN in cancer, we could show that not only replication fork progression is affected, but also fork stalling and origin firing. ATR safeguards stalled replication forks from collapse and prevents replication catastrophe [36]. ATR kinase activity slows down replication forks and prevents origin firing in damaged cells. By suppressing the excessive firing of replication origins, this prevents the exhaustion of the rate-limiting pool of RPA and the conversion of ssDNA to DSBs in the S phase, a serious threat to genome stability. Our results demonstrate that considerable effort needs to be expended to identify additional synthetic lethal interactions with genes of the replication stress response.

PARP1 inhibitors have emerged as promising cancer therapeutics, especially for BRCA-mutated tumors that display a deficiency in HR [15]. An ongoing phase II trial of the PARP1 inhibitor Talazoparib is testing patients with pathologic mutations in a somatic or germline non-BRCA1/2 HR pathway gene, such as *PTEN* (NCT02401347). However, further data on the use of PARP1 inhibitors in non-BRCA HR deficiencies are still missing [16]. Previously, we reported that PTEN contributes to HR in prostate cancer and proposed the treatment of agents targeted against defects in HR such as PARP1 inhibitors to PTEN-deficient tumors [9]. Here, we show for the first time that TNBC cell lines with low PTEN protein levels can be sensitized for treatment with PARP1 inhibitors, independent of BRCA mutations or a BRCA-like phenotype. Only 10% or less of the cells survived the treatment with 10 µM Olaparib, which is comparable with the survival rate found in BRCA2 knock out cells or BRCA deficient cell lines [37].

Cells have acquired a multitude of fork protection mechanisms to minimize the genotoxic effects during replication stress [38]. The regulation of replication stress in response to DNA damage involves the reduction in fork progression, the regulation of origin firing [39] and the stabilization of stalled forks [40]. There is evidence that PTEN is an important factor for the stabilization of replication forks, and thus maintains genome stability under replication stress [41]. Wang et al. showed that PTEN is located at replication sites, and physically interacts with replication protein A 1 (RPA1) as well as Ubiquitin Thioesterase OTUB1, which regulates RPA1 stability. After HU-treatment, the length of nascent replication tracts was decreased in PTEN -/- HCT116. Our data support the observation that PTEN also plays an important role in protecting replication forks in TNBC by showing a reduction in CldU track length after HU treatment in cells with lower amounts of PTEN (Figure 5). Here, the amount of degraded DNA negatively correlated with the amount of PTEN. This implies that the amount of PTEN could be essential for fork protection. By analyzing SKBR-3, MCF7, T47D and BT474 we could show that PTEN is a general protective factor at stalled replication forks (Appendix A). There is also evidence that loss of PTEN induces uncontrolled origin firing (Figure 3g). Replication fork stalling and degradation of newly replicated DNA results in regions of ssDNA. ssDNA is protected by RPA coating, followed by the recruitment and activation of ATR and phosphorylation of its main downstream effector CHK1 [42]. The phosphorylation of CHK1 stabilizes the forks and downregulates origin firing, thus delaying the cell cycle progression and allowing to repair DNA damage. Dysregulation in this process may give rise to uncontrolled initiation of origin firing and fork collapse, thus causing genomic instability.

Latterly, we analyzed the effect of different amounts of PTEN at the chromatin using MDA-MB-231 BR to overexpress PTEN, and compared it to moderate expression of PTEN (MDA-MB-231) and no expression of PTEN (BT549) (Figure 6). We could show that PTEN is localized to the chromatin in untreated cells. Further, the amount of PALB2, PARP1 and CHK1 at the chromatin was dependent on the amount of PTEN (Figure 6a). It has been reported for many HR proteins, like BRCA1 and BRCA2, that they might have a repair independent role of protecting stalled replication forks [43,44]. PARP1 has also been implicated in the restart of stalled replication forks through a mechanism that is dependent on Meiotic Recombination 11 Homolog A (Mre11), suggesting that DNA end resection is a critical control point regulated by PARPs [45]. Further, CHK1 has been suggested to bind the PAR chain synthesized by PARP1 at stalled replication forks, which facilitates its kinase activity [46]. Besides its role in stalled replication forks, CHK1, together with ATR, is a regulator of the S-phase checkpoint and the global regulation of further origin firing and activation of DNA repair upon replication stress [39]. After HU treatment, we found an almost two-fold lower phosphorylation of CHK1 for BT549 compared to MDA231/PTEN and MDA231. Interestingly, the recruitment and activation of ATR was not affected (Figure 6b). Due to the great impact of ATR in the regulation of stalled replication forks and origin firing, we think that ATR might also be a good target in the treatment of TNBC with high CIN [47].

Finally, the comparison of replication processes in TNBC with low amounts of PTEN and PTEN overexpression provides strong evidence that the amount of PTEN is crucial for replication processes (Appendix A).

We conclude from our data that PTEN might be an important regulator of replication proteins to the chromatin to protect and regulate the action of replication processes. However, the exact process behind this needs further investigation.

With our work, we could offer detailed insight into the importance of PTEN in the maintenance of CIN by regulating DNA replication processes. We consider that the high CIN of PTEN-deficient TNBC predestines it for treatment with PARP1 inhibitors due to the protective role of PTEN during DNA replication.

## 4. Material and Methods

### 4.1. Cell Culture, Plasmids and Survival Assays

The breast cancer cell lines BT20, HS578T, GI101, MCF7, MDA231, MDA468 and SKBR3 were cultivated in DMEM and BT474, BT549 and T47D (provided by the Department of Tumor Biology) in RPMI media, both supplemented with 10% FCS, 2 mM glutamine, 100 U/mL penicillin and 100 mg/mL streptomycin at 37 °C and 10% CO_2_ or 5% CO_2_ respectively. Zsgreen-positive MDA-MB-231 BR control (231BR/CTL) and PTEN-overexpressing (MDA231/PTEN) cells were generated by lentiviral transduction with either control LeGO-SWITCH vector piZs2puro++tTR+ or overexpression vector pPTENiZs2puro++tTR+ [25] and cultured in DMEM supplemented with 1 μg/mL doxycycline. Stable transfectants were selected in DMEM supplemented with 4 μg/mL puromycin. Suitable concentrations of doxycycline as well as puromycin were determined by titration.

For survival assays, 250 cells were seeded in a 6-well plate 6 h before treatment and cells were cultured for 14 days. Cells were fixed and stained with 1% crystal violet (Sigma-Aldrich, St. Louis, MO, USA). Colonies with more than 50 cells were counted and normalized to untreated samples. Each survival curve represents the mean of at least three independent experiments.

### 4.2. Clinical in Silico Analysis

Gene expression data for *PTEN* and CIN70 genes in the METABRIC dataset (PMID: 27161491) was retrieved from the cBioPortal (PMID: 22588877, PMID: 23550210). For each tumour, the CIN70 score was calculated according to Birkbak et al. (PMID: 21270108) by summation of expression values of all the CIN70 genes. Afterwards, *PTEN* gene expression was compared in the extreme quartiles of CIN70 score. *p*-values were calculated by Student’s *t*-test.

### 4.3. Mutagenesis Assay

PARP1 inhibition was achieved by adding 1 µM Olaparib (Selleck, Houston, TX, USA) for 90 min to the growth medium. For fork stability experiments, cells were treated with 2 mM Hydroxyurea (HU) (Sigma-Aldrich, Darmstadt, Germany) added to the growth medium for four hours. All treatments were performed at 37 °C and in 10% CO_2_ or 5% CO_2_ atmosphere.

### 4.4. Western Blotting

Proteins were extracted and concentration was determined via photometry. A total of 40 µg were resolved by SDS-PAGE using a gradient gel [(4–15%), Bio-Rad Laboratories, Feldkirchen, Germany].

Fractionated extracts were generated using the “Subcellular Protein Fractionation Kit for cultured cells” (Thermo Scientific, Waltham, MA, USA), following the Manufacturer’s instruction. After harvesting the cells, all incubation and centrifugation was performed at 4 °C. For each step, phosphatase inhibitor (Halt™ Phosphatase Inhibitor Cocktail, Thermo Scientific, Waltham, MA, USA) was added in addition to the protease inhibitor cocktail included in the Kit. The cytoplasmatic, nuclear and chromatin-bound fractions were isolated, concentration was determined by photometry, and 20 μg were resolved by SDS-PAGE using a gradient gel [(4–15%), Bio-Rad Laboratories, Feldkirchen, Germany].

After transfer to a Nitrocellulose membrane (Licor, Lincoln, NE, USA), proteins were detected by anti-PTEN IgG (9559, 1:2000), anti-Chk1 IgG (2360, 1:750), anti-p-Chk1 IgG (2348, 1:1000), anti-p-ATR (2853, 1:1000) (Cell Signaling Technology, Danvers, MA, USA), anti-ATR IgG (SC1887, 1:1000), anti-PALB2 IgG (A301-2464, 1:1000) and anti-GAPDH IgG (SC25778, 1:1000) (Santa Cruz Biotechnology, Dallas, TX, USA), anti-PARP1 IgG (556362, BD Bioscience, San Jose, CA, USA, 1:1000), anti-H2B IgG (NB100-56347, Novus Biologycals, Littleton, CO, USA, 1:500)or anti-β-actin IgG (Sigma, Darmstadt, Germany, 1:50.000), IRDYE 680 conjugated anti-mouse IgG or IRDYE 800 conjugated anti-rabbit IgG (Licor, Lincoln, NE, USA, 1:7500).

### 4.5. DNA Fiber Assay

Exponentially growing cells were pulse-labeled with 25 μM CldU (Sigma, Darmstadt, Germany) and 250 μM IdU (Sigma, Darmstadt, Germany) added to the growth medium for the times specified. Where indicated, the cells were exposed to 2 mM HU or 1 µM Olaparib during or in between the pulse labeling, as indicated. Labeled cells were harvested and DNA fiber spreads were prepared from 0.5 × 10^6^ cells/mL as described previously [12]. Slides were incubated in 2.5 M HCl for 80 min and then washed three to five times in PBS, followed by incubation in blocking buffer (2% BSA, 0.1% Tween in PBS) for 1 h. Acid-treated fiber spreads were stained with monoclonal rat anti-BrdU antibody (Bio-Rad, Feldkirchen, Germany) (1:1000) to detect CldU, and monoclonal mouse anti-BrdU antibody (Becton Dickinson, Franklin Lakes, NJ, USA, 1:1500) to detect IdU. Secondary antibodies were goat anti-rat AlexaFluor555 and goat anti-mouse AlexaFluor488 (both Invitrogen, Carlsbad, CA, USA, 1:500). Primary antibodies were diluted in blocking buffer, incubated for 1 h with rat anti-BrdU antibody and mouse anti-BrdU antibody, followed by fixation with 4% paraformaldehyde and extensive washes in PBS and blocking buffer. Secondary antibodies were applied for 1.5 h, washed with PBS and blocking buffer and slides were mounted in Immuno-Fluor mounting medium (MP Biomedicals, Santa Ana, CA, USA). Fiber tracts were examined using an AxioVert 200 M fluorescence microscope (Zeiss, Oberkochen, Germany). Pictures were taken from randomly selected fields with untangled fibers and analyzed using ImageJ software package. For structure analyses, the frequencies of the different classes of fiber tracks were classified as follows: red-green (elongating fork), red (stalled or terminated forks), green-red-red-green (first pulse origin) and green (second pulse origin). At least 300 forks were analyzed for each experiment and the means of at least three independent experiments are represented. For fork speed analyses, the lengths of CldU and IdU tracks were measured and micrometer values were converted into kilobases. A conversion factor for the length of a labeled track of 1 μm = 2.59 kb was used [13]. A minimum of 100 individual fibers were analyzed for each experiment and the means of at least three independent experiments are presented.

### 4.6. Data Evaluation

Statistical analysis, curve fitting and graphs were performed by means of the Prism software Version 6 (Graph Pad Software, San Diego, CA, USA). Data are given as mean (±SEM) of replicate experiments. 

## 5. Conclusions

Impaired DNA replication occurs in cancer, where it contributes to genomic instability. This vulnerable process is already a target of cancer therapies. Drugs that increase replication stress have become attractive for therapeutic intervention in several cancer types. Our results suggest that the clinical use of PARP1 inhibitors should be extended beyond those tumors with BRCA mutations to a larger group of patients with PTEN mutant tumors, maybe even to the larger group of CIN high tumors.

## Figures and Tables

**Figure 1 cancers-12-02809-f001:**
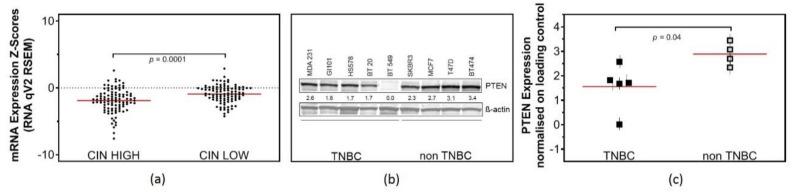
High chromosomal instability (CIN) scores and low phosphatase and tensin homolog (PTEN) expression in triple-negative breast cancer (TNBC). (**a**) In silico analysis of PTEN mRNA expression data, comparing 100 TNBC patients with a high CIN70 score to 100 TNBC patients with a low CIN70 score. (Student’s *t*-test *p* = 0.0001) (**b**) Western Blot analysis of PTEN expression in TNBC (MDA231, BT549, BT20, GI101 and HS578T) and non-TNBC breast cancer cell lines (MCF7, T47D, SKBR3, BT474) in relation to ß-actin as loading control. (**c**) Grouped analysis of quantitative PTEN expression normalized on ß-actin, comparing TNBC and non-TNBC breast cancer cell lines. (Student’s t-test *p* = 0. 0416). Each value represents the mean of the quantitative analysis of three independent experiments with the SEM indicated.

**Figure 2 cancers-12-02809-f002:**
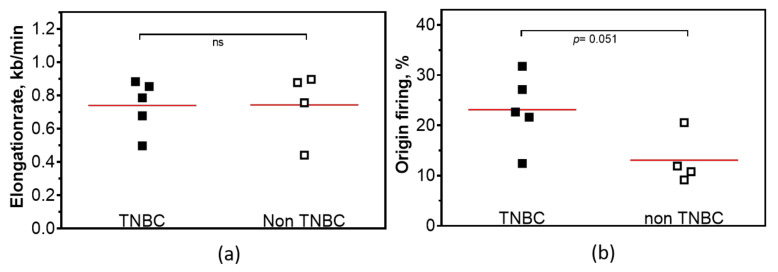
Replication stress in TNBC compared to non-TNBC. (**a**) Replication fork progression and (**b**) activation of replication origins in untreated cell lines of TNBC in comparison to untreated non-TNBC breast cancer cell lines. (Student’s *t*-test *p* = 0.98 (**a**) and *p* = 0.051 (**b**). Each value represents the mean of three independent experiments.

**Figure 3 cancers-12-02809-f003:**
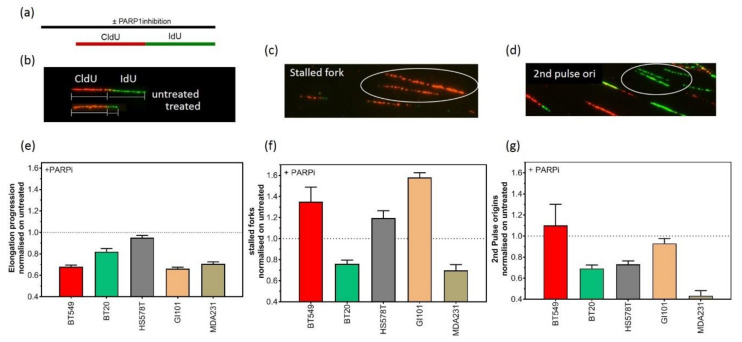
Replication fork elongation, stalling and new origin firing after PARP1 inhibition. (**a**) Overview of DNA fiber assay labelling protocol. Representative picture of reduced IdU-fiber length after treatment, in comparison to full-length IdU fiber in untreated sample (**b**), of CldU labelled, IdU-negative fiber indicating stalled replication fork (**c**) and representative picture of IdU labelled, CldU negative fiber, indicating second pulse origin firing (**d**). Quantitative analysis of elongation tract length (**e**), replication fork stalling (**f**) and second pulse origin firing (**g**) after PARP1 inhibition normalized on untreated controls. Each value represents the mean of at least three independent experiments with SEM indicated.

**Figure 4 cancers-12-02809-f004:**
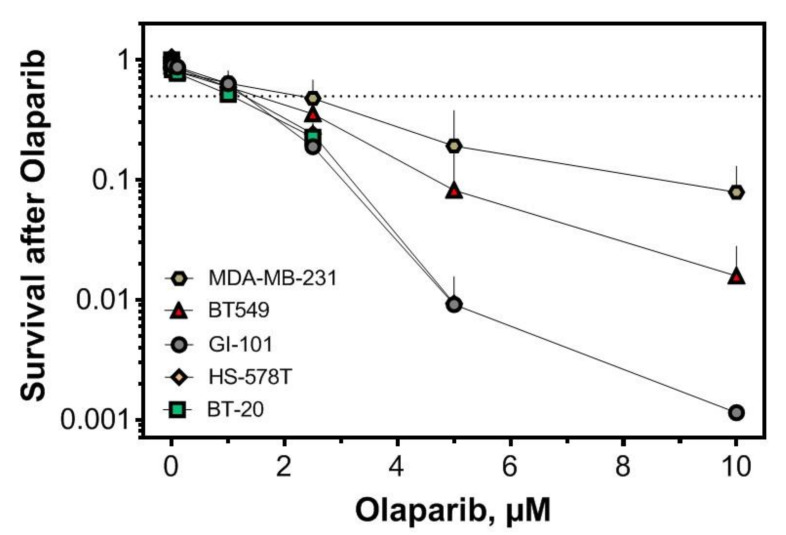
Cellular sensitivity of TNBC to PARP1 inhibition. Colony-forming assays were performed after treatment, with increasing concentrations of the PARP1 inhibitor Olaparib for 90 min. Each value represents the mean of at least three independent experiments, with SEM indicated.

**Figure 5 cancers-12-02809-f005:**
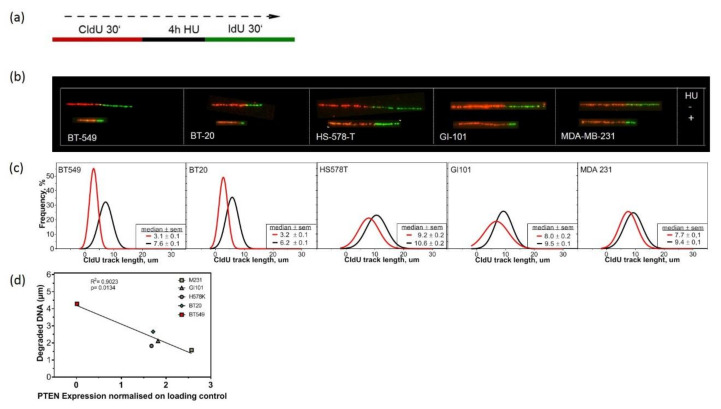
Fork instability after Hydroxyurea (HU) treatment in TNBC, dependent on PTEN expression. (**a**) Overview of DNA fiber assay labelling protocol. (**b**) Representative images of DNA fibers of 5 TNBC cell lines after treatment with 2 mM HU for four hours and untreated cells. Unstable replication fork indicated by shortening in CldU-labelled tract length in comparison to untreated controls. (**c**) Quantitative analysis of CldU elongation track length after treatment with 2 mM HU in comparison to untreated controls. (red: HU treatment, black: untreated) (**d**) CldU elongation length correlates negatively with PTEN expression measured at protein level (see Figure 1). Each value represents the mean of three independent experiments with the SEM indicated.

**Figure 6 cancers-12-02809-f006:**
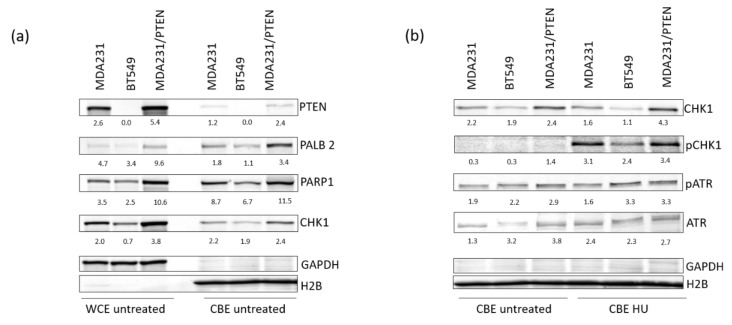
PTEN expression and recruitment of DNA repair proteins to the chromatin in TNBC. (**a**) Western blot analysis of PTEN, PALB2, PARP1 and CHK1 expression in whole cell extracts (WCE) and chromatin-bound extracts (CBE) of untreated TNBC cell lines. GAPDH and H2B served as a loading control for the WCE and CBE, respectively. (**b**) Western blot analysis of CHK1, pCHK1, ATR and pATR expression at the chromatin after treatment with 2 mM HU for 4 h in comparison to untreated controls. H2B served as a loading control.

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
