# Peer review of "Exploiting Chromosomal Instability of PTEN-Deficient Triple-Negative Breast Cancer Cell Lines for the Sensitization Against PARP1 Inhibition in a Replication-Dependent Manner"

_cancers, 2020, doi:10.3390/cancers12102809_

Round 1
Reviewer 1 Report
I am satisfied that the authors have been using a variety of cell lines as a model system to elucidate the role of PTEN. It is a wonderfully simple and reproducible system that can be genetically modified and thus may serve as a first step in exploring putative targets for treatment.
The authors may seek advice on the unusual use of some of the terms such as "recognise" (e.g. lines 22, 31), "vulnerability" (line 20), "latterly" (l. 251), "think" (l. 266) - although the chosing of these words are not disturbing...
l. 313: "were" instead of "where"
l. 271-272: Should be rephrased
Author Response
We would like to thank the reviewer for their valuable time and useful contribution. We appreciate the effort the reviewers have given. We hope that we have addressed all the issues raised by the reviewers. At the end, we are convinced that the reviewers’ suggestions helped to improve our manuscript and hope that the manuscript now clearly demonstrates the importance of the presented study for TNBC.
We corrected "where" to "were" (Line 354).
Reviewer 2 Report
In the present manuscript, Rieckhoff et al. found that PTEN plays an important role in chromosomal instability on TNBC cells. They also suggested that TNBC cells with low PTEN expression are more sensitive to PARP1 inhibitor, and these phenomena are independent on BRCA mutations or BRCA-like phenotype. In addition, they also identified that PTEN plays important roles in regulating replication processes.
However, there are several questions are still needed to be clarified.
- Many studies (such as Curr Drug Targets. 2014 Jan; 15(1): 65–79., EMBO Mol Med. 2009 Sep;1(6-7):315-22., Cancer J. 2010 Jan-Feb; 16(1): 48–52.) have indicated that PTEN plays important in chromosomal instability and DNA repair, and PARP1 inhibitor could induce the synthetic lethal effect in PTEN mutation/loss of function cancer cells. The authors should discuss the differences and novelty between their finding and other investigators’.
- BT549 expressed the lowest expression of PTEN in these TNBC/non TNBC cells. Why BT549 is not more sensitive to Olaparib than other TNBC cells?
- Why authors use MDA-MB-231 as cell model in PTEN overexpression experiments? The basal expression of PTEN in MDA-MB-231 is higher than other TNBC cells. Maybe author could use BT-20 to do the same experiments.
- In addition, why authors didn’t do the experiments to compare the cytotoxicity induce by Olaparib between normal and PTEN knockdown cells?
Reviewer 3 Report
See enclosed file

Round 2
Reviewer 2 Report
The authors have made satisfactory improvements to the manuscript based on the issues that were raised.
Reviewer 3 Report
The authors have addressed all my concerns and comments. They did an excellent job at this.